# On parsimony and clustering

Frédérique Oggier[1] and Anwitaman Datta[2]

[1] School of Physical & Mathematical Sciences, Nanyang Technological University Singapore, Singapore
[2] School of Computer Science & Engineering, Nanyang Technological University Singapore, Singapore

## ABSTRACT

This work is motivated by applications of parsimonious cladograms for the purpose of analyzing non-biological data. Parsimonious cladograms were introduced as a means to help understanding the tree of life, and are now used in fields related to biological sciences at large, *e.g.*, to analyze viruses or to predict the structure of proteins. We revisit parsimonious cladograms through the lens of clustering and compare cladograms optimized for parsimony with dendograms obtained from single linkage hierarchical clustering. We show that despite similarities in both approaches, there exist datasets whose clustering dendogram is incompatible with parsimony optimization. Furthermore, we provide numerical examples to compare *via* F-scores the clustering obtained through both parsimonious cladograms and single linkage hierarchical dendograms.

## INTRODUCTION

Systematics is a field of biology that seeks to reconstruct the evolutionary history (phylogeny) of life, namely how different kinds of organisms on earth evolved to become the currently known diversity of species (*Lipscomb, 1998*). Phylogeny is usually represented by a leaf-labeled tree where the non-leaf nodes refer to (possibly hypothetical) ancestors and the leaves are labeled by the species. A phylogenetic tree is also known as a cladogram. Apart from understanding the history of life, phylogeny has other applications (*Sung, 2009*), for instance, to analyze rapidly mutating viruses (such as the HIV virus, or the SARS-CoV-2 virus, for example, *Morel et al. (2020)*, *Thornlow et al. (2022)* and *Hadfield et al. (2018)*, *National Institutes of Health (2023)* for datasets), for multiple sequence alignment (alignment of three or more biological sequences, protein or nucleic acid, of similar length), for the prediction of the structure of proteins and ribonucleic acid (RNA), to predict gene expression and ligand structures for drug design.

The problem of phylogenetic tree reconstruction is to predict or estimate the phylogeny for some input data. Two types of methods can be used: (1) character-based methods (maximum parsimony (for example, *Robinson (1971)* and maximum likelihood), where character refers to any observable feature or trait of an organism, (2) distance-based methods (for example, unweighted pair group with arithmetic mean (for example, *Saitou & Nei (1987)*, transformed distance and neighbour relation). In this work we focus on the parsimony approach, and use the term parsimonious cladograms to refer to cladograms considered with the parsimony criterion.

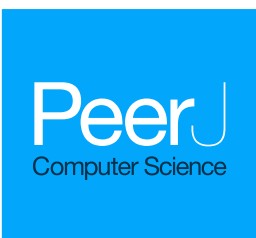

Corresponding author
Anwitaman Datta,
anwitaman@ntu.edu.sg

The notion of parsimony can be traced back to *Hennig (1950)* and *Edwards & Cavalli-Sforza (1963)*; see *Farris & Kluge (1997)* for a historical perspective around the notion of parsimony. Trees following the principle of parsimony minimize the number of changes from one node to another. In other words, parsimony states that the simplest explanation that explains the greatest number of observations is the preferred one. Different variants of parsimony exist, depending on properties that characters may have. Characters are reversible if changes may happen several times (otherwise, each character changes at most once). Characters are ordered if changes have to happen in a particular order.

In this work, we consider only Fitch's parsimony (*Fitch, 1971*), where characters are unordered, multistate, and reversible. Other variations include perfect phylogeny, Wagner, Dollo and Camin-Sokal parsimony (see chapter 4 of *Kitching et al. (1998)*), which are outside the scope of this initial study.

Trying to best group objects based on their shared properties is the goal of clustering. Since there is an enormous amount of literature on clustering (see for example, *Xu & Tian (2015)* and *Saxena et al. (2017)*, we will narrow our discussion to agglomerative hierarchical clustering: given points to be clustered, a distance is used to compare them; an initial grouping is performed, the process is then iterative, and at each iteration, existing groups are agglomerated until there is a single cluster. The resulting clustering can be visualized *via* a dendogram (while the term dendogram is also used in the context of parsimony, we will use cladogram in the context of parsimony, and dendogram in the context of clustering). Hierarchical clustering is a well known classical technique (see *Nielsen (2016)*, *Defays (1977)*, *Sibson (1973)*). The reason for which we consider it, rather than say a more modern approach, is because the variant called 'single linkage' (*Sibson, 1973*) resembles closely with the parsimony method that we explore.

The question addressed in this work pertains to the usage of parsimony as a clustering measure, and could be phrased as: *what are the congruences, if any, between the cladograms obtained using Fitch parsimony as an optimization criterion, and dendograms obtained from agglomerative hierarchical clustering?* It is a partial question to a more ambitious one: *what are the connections, if any, between the cladograms obtained using maximum parsimony as an optimization criterion, and dendograms obtained from agglomerative hierarchical clustering?* The relations between evolution on the one hand and discoverability of characters, hierarchy and parsimony on the other hand were addressed in *Brower (2000)*. This question is motivated by the use of maximum parsimony cladograms for analyzing non-biological, and more more generally, non-ancestry based data (for instance, volcanoes in *Hone et al. (2007)*). If agglomerative clustering could lead to good enough parsimonious trees for (very) large datasets, this would provide computationally efficient approximation algorithms. Computing parsimonious trees is not as computationally efficient as hierarchical clustering; the latter has been carried out with up to 100 gigabytes of data (*Sun et al., 2009*). If on the other hand, parsimonious trees cluster data in a unique manner, it would be meaningful to compare what features of the data are being captured with parsimony compared with respect to those captured by a standard clustering algorithm. This could prove useful for non-biological (not necessarily

---

**Algorithm 1:** An algorithm to compute a parsimonious tree from a minimum spanning tree.

**Data:** $T_1$ a minimum spanning tree, with edge set $E(T_1)$

**Result:** $T_2$ a labeled tree, with edge set $E(T_2)$

$E(T_2) \leftarrow \varnothing$;

**while** $|E(T_1)| \neq 0$ **do**

    $L \leftarrow leaves(T_1)$;

    $P \leftarrow predecessors(L)$;

    **if** $P \neq L$ **then**

        $E(T_2) \leftarrow E(T_2) + \{(u,v,d(u,v)),\ u \in L, v \in P\}$;

        $E(T_2) \leftarrow E(T_2) + \{(v,v,0),\ v \in P\}$;

        $E(T_1) \leftarrow E(T_1) - \{(u,v,d(u,v)),\ u \in L, v \in P\}$;

    **else**

        $E(T_2) \leftarrow E(T_2) + \{(L[0], L[1], d(L[0], L[1]))\}$;

        $E(T_2) \leftarrow E(T_2) + \{(L[1], L[1], 0)\}$;

        $E(T_1) \leftarrow E(T_1) - \{(L[0], L[1], d(L[0], L[1])),\ \}$;

    **end**

**end**

---

ancestry-based) applications, with not too large datasets. While this work certainly does not contain all the answers, it provides a first insight into these questions.

Recent works have studied parsimonious trees and clustering, but in different contexts. In *Mawhorter & Libeskind-Hadas (2019)*, hierarchical clustering is used for the purpose of phylogenetic reconciliation, which consists of inserting a phylogenetic tree representing the evolution of an entity into the phylogenetic tree of an encompassing entity to reveal some possible shared history. In *Brucker & Gély (2009)*, a new clustering structure, named parsimonious cluster systems, is introduced as a generalization of phylogenetic trees.

In "Parsimonious cladograms", we formalize the optimization involved in finding parsimonious cladograms, and given a set $S$ of sequences as input data, we provide a simple algorithm (Algorithm 1) to compute a cladogram for $S$ whose parsimony is the weight of a minimum spanning tree computed from $S$. In "Hierarchical clustering and dendograms", we recall the principles behind single linkage hierarchical clustering, and propose a second algorithm (Algorithm 3), which, given a set $S$ of sequences to be clustered, outputs a dendogram which also forms a parsimonious cladogram. The combination of both algorithms enables us to compare parsimonious cladograms with guaranteed parsimony with single linkage dendograms. We exhibit examples of datasets for which the parsimony achieved by Algorithm 1 cannot possibly be achieved by a single linkage dendogram, even though Algorithm 3 provides a single linkage dendogram but with worse parsimony. Numerical experiments are provided in "Numerical exploration". Four methods to construct parsimonious trees are compared (*via* F-score) in terms of the clusterings they

---

**Algorithm 2:** A standard agglomerative hierarchical clustering.

**Data:** $n$ points $P_1, \ldots, P_n$

**Result:** A hierarchical clustering of $P_1, \ldots, P_n$

$C_i \leftarrow P_i, \; i = 1, \ldots, n;$

$C \leftarrow \{C_1, \ldots, C_n\};$

**while** $|C| > 1$ **do**

    $D_{ij} \leftarrow d_{BC}(C_i, C_j)$ for $C_i, C_j \in C;$

    $\hat{C}_i, \hat{C}_j \leftarrow \arg\min D_{ij};$

    $C \leftarrow C + \hat{C}_i \cup \hat{C}_j;$

    $C \leftarrow C - \{\hat{C}_i, \hat{C}_j\};$

**end**

---

**Algorithm 3:** A modified agglomerative hierarchical clustering.

**Data:** $n$ points $P_1, \ldots, P_n$

**Result:** A parsimonious tree $T$ for $P_1, \ldots, P_n$

$C_i \leftarrow P_i, \; i = 1, \ldots, n;$

$E(T) \leftarrow \varnothing;$

$labels \leftarrow \varnothing;$

$C \leftarrow \{C_1, \ldots, C_n\};$

**while** $|C| > 1$ **do**

    $D_{ij} \leftarrow d_{BC}(C_i, C_j)$ for $C_i, C_j \in C;$

    $\hat{C}_i, \hat{C}_j \leftarrow \arg\min_{C_i, C_j} D_{ij};$

    $\hat{P}_i, \hat{P}_j \leftarrow \arg\min_{x \in \hat{C}_i, y \in \hat{C}_j} d_H(x, y);$

    $C \leftarrow C + \hat{C}_i \cup \hat{C}_j;$

    $E(T) \leftarrow E(T) + \{(\hat{C}_i, \hat{C}_i \cup \hat{C}_j), (\hat{C}_j, \hat{C}_i \cup \hat{C}_j)\};$

    **if** $\hat{C}_i$ is a leaf **then**

        $label(\hat{C}_i) \leftarrow P_i;$

    **else**

        $label(\hat{C}_i) \leftarrow \hat{P}_i;$

    **end**

    **if** $\hat{C}_j$ is a leaf **then**

        $label(\hat{C}_j) \leftarrow P_j;$

    **else**

        $label(\hat{C}_j) \leftarrow \hat{P}_j;$

    **end**

    $labels \leftarrow labels + \{label(\hat{C}_i), label(\hat{C}_j);$

    $C \leftarrow C - \{\hat{C}_i, \hat{C}_j\};$

**end**

---

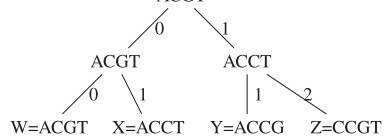

| | properties | | | |
|---|---|---|---|---|
| W | A | C | G | T |
| X | A | C | C | T |
| Y | A | C | C | G |
| Z | C | C | G | T |

**(a)** A set $S$ of sequences, with $S = \{$ACGT, ACCT, ACCG, CCGT$\}$ .

**(b)** A tree $T$ for the set $S$, with a labeling such that $w(T) = 5$.

**Figure 1** **An example of tree (from** *Sung (2009))* **for the sequence set** $S$ **= {ACGT, ACCT, ACCG, CCGT} using the Hamming distance.**

yield. These show divergent patterns between parsimony and tree structures, in the sense that having a parsimony that decreases does not imply tree structures and thus clusters becoming more similar.

# PARSIMONIOUS CLADOGRAMS

## Formal definitions

A tree $T$ is an undirected graph with no cycle, with vertex set $V(T)$ and edge set $E(T)$. We will consider rooted trees.

Let $\mathscr{S}^n$ be the set of all possible sequences (not necessarily observed) of length $n$ over a finite alphabet. For example, the alphabet may be binary to state the presence or absence of a trait, or multistate, for instance, A, C, T, G for a multiple sequence alignment.

We need a distance function on sequences $x, y \in \mathscr{S}^n$. Recall that a distance function $d$ satisfies the following axioms: (i) $d(x, y) \geq 0$ and $d(x, y) = 0 \iff x = y$, (ii) $d(x, y) = d(y, x)$ and (iii) the triangle inequality: $d(x, y) + d(y, z) \geq d(x, z)$, for all $x, y \in \mathscr{S}^n$. Recall also that we consider only the case of unordered characters in this work.

**Definition 1.** Let $d$ be a distance function. Let $T$ be a rooted tree whose leaves are a subset $S$ of sequences in $\mathscr{S}^n$, and each non-leaf node is assigned a sequence in $\mathscr{S}^n$. The weight or parsimony length of $T$ is defined by

$$w(T) = \sum_{(x,y) \in E(T)} d(x, y)$$

where $d(x, y)$ denotes the distance between $x$ and $y$, and we identify nodes in the graph with their sequence. Given a set $S$, the most parsimonious tree (or most parsimonious cladogram) is a tree $T^*$ whose parsimony length $w(T^*)$ is minimized over all possible trees whose leaves are labeled by $S$ and all possible sequence assignments of the non-leaf nodes. We note that any tree with label has an associated parsimony. As such, there is a certain abuse of the nomenclature 'parsimonious tree' or 'parsimonious cladogram', with which we refer to any tree with a label that may however not be optimal. This phrasing nonetheless allows us to refer to such trees succinctly and is also consistent with the notion of most (or maximum) parsimonious tree.

Let $d_H(x, y)$ be the Hamming distance between two sequences $x \neq y \in \mathscr{S}^n$. The Hamming distance counts the number of positions in which $x, y$ differ. In Fig. 1, a tree for the sequence set $S$ = {ACGT, ACCT, ACCG, CCGT} is shown (see Fig. 1B), where $d = d_H$. The tree has $S$ for leaves, and the choice of labels for the non-leaf nodes gives a parsimony

length $w(T) = 5$. In Fig. 1A, the objects or species $W$, $X$, $Y$, $Z$ are identified with the sequences representing their properties or characters.

There are two standard problems associated to parsimony (Sung, 2009):

- The 'small parsimony' problem: it consists of finding the optimal labels for internal nodes given a tree with labelled leaves. This has been solved in Fitch (1971).
- The 'large parsimony' problem: it refers to the tree reconstruction problem, namely, given a set $S$ of sequences, to find the most parsimonious tree. Since this problem is NP-complete (Sung, 2009, 7.2.1.2), in practice, approximation algorithms to identify good (if not optimal) solutions are used.

**Large parsimony**

Let $S$ be a set of sequences. Let $G(S)$ be the complete graph with vertex set $S$. Every edge $(x, y)$ has weight $d(x, y)$, where $x, y$ are sequences in $S$, identified with their nodes. Let $T$ be the minimum spanning tree of $G(S)$, and let $T^*$ be the most parsimonious tree given $S$. The following bound on the parsimony length of a tree $T$ is known (Sung, 2009, Lemma 7.2.).

**Proposition 1.** *Let* T *be a minimum spanning tree of $G(S)$, and let $T^*$ be the most parsimonious tree for S. Then*

$$w(T) \leq 2w(T^*).$$

*Proof.* Let $C$ be an Euler tour of $T^*$. Without loss of generality, assume that the tour starts at the root. It will then traverse $T^*$ in such a way that each vertex is added to the tour when it is visited (either moving down from a parent vertex or returning from a child vertex). The tour returns to the starting node after visiting all the vertices of $T^*$, in the process traversing every edge twice (and revisiting nodes in the process). Thus

$$w(C) = 2w(T^*).$$

Consider now a walk on $G(S)$ which visits the vertices in the same order in which they appear in $C$, and return to the start node at the end. Then, since the distance function satisfies the triangle inequality, the length of this walk is less than or equal to $w(C)$. Now, if we remove any edge from this walk, it creates a shorter walk $P$, which is in turn also shorter than or equal to the length of the original walk. Consequently, we have

$$w(P) \leq w(C).$$

Finally, since $P$ is a superset of a spanning tree, we have

$$w(T) \leq w(P) \leq w(C) = 2w(T^*)$$

for $T$ is a minimal spanning tree.

There are two ways to interpret this inequality. Foremost, the optimal parsimony tree weight $w(T^*)$ is at least half the weight of the minimum spanning tree. Conversely, the minimum spanning tree can be harnessed to design a two-approximation polynomial time algorithm for parsimonious tree construction. Furthermore, while the inequality

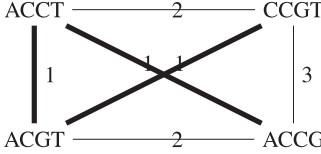

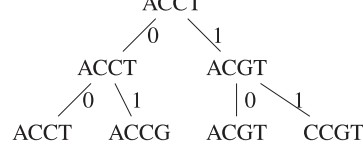

**(a)** The complete graph $G(S)$ with its minimum spanning tree.

**(b)** A tree $T$ for $S$, with weight lower than that of Figure 1b.

**Figure 2** For the set $S$ = {ACGT, ACCT, ACCG, CCGT}, its complete graph $G(S)$ whose edges have for weight the Hamming distance between the nodes they connect, a minimum spanning tree of $G(S)$ and the resulting cladogram.

guarantees a tree so constructed to have a parsimony no worse than twice that of the most parsimonious tree, a keen inspection of the last line of the proof of the inequality suggests that in practice this would generally yield significantly better parsimonious tree instances. We note that better than two-approximation polynomial algorithms have been proposed, for instance, *Alon et al., (2008)* achieving 1.55-approximation asymptotically. More recently, *Jones, Kelk & Stougie (2021)* showed that the maximum parsimony problem is fixed parameter tractable, in that deciding whether the maximum parsimony is greater or equal to a fixed threshold can be solved in polynomial time.

Consider the example in Fig. 2, with $d$ the Hamming distance, the minimum spanning tree $T$ shown on Fig. 2A has weight $w(T) = 3 \le 2w(T^*)$, so we know that $w(T^*) \ge 1.5$. A tree with weight 3 is also shown on Fig. 2B, which improves on the tree from Fig. 1.

In Algorithm 1 we explicit how a parsimonious tree $T_2$ can be derived from a minimum spanning tree $T_1$ of the complete graph $G(S)$, such that $w(T_1) = w(T_2)$. Combining Proposition 1 and Algorithm 1 gives

$$\frac{1}{2}w(T) \le w(T^*) \le w(T)$$

where $T^*$ is the most parsimonious tree for $S$ and $T$ is a minimum spanning tree for $G(S)$.

The algorithm starts with the leaves of $T_1$, which becomes leaves of $T_2$, and furthermore, for each predecessor $p$ in the set of predecessors of these leaves, an edge $(p, p)$ of weight 0 is added. Since both leaves of $T_1$ and non-leaves of $T_1$ all become leaves in $T_2$, and by definition of minimum spanning tree, this means all and only elements of $S$ are leaves in $T_2$, $T_2$ is indeed a parsimonious tree, with the specificity that the labels of intermediate nodes are in $S$. Since the edge set $E(T_2)$ of $T_2$ is built from that of $E(T_1)$, apart from the edges of weight 0 which do not contribute to the weight, we have $w(T_1) = w(T_2)$.

Algorithm 1 does not optimize labelings for the constructed tree, it merely exhibits a tree whose weight is guaranteed to be that of the minimum spanning tree, which we will use later on in this work. Once the tree is formed, the labeling can be optimzed using (*Fitch, 1971*).

## HIERARCHICAL CLUSTERING AND DENDOGRAMS

Agglomerative hierarchical clustering clusters $n$ points $P_1, \ldots, P_n$ iteratively, starting from individual points up to having one cluster with all points (by contrast, divisive hierarchical

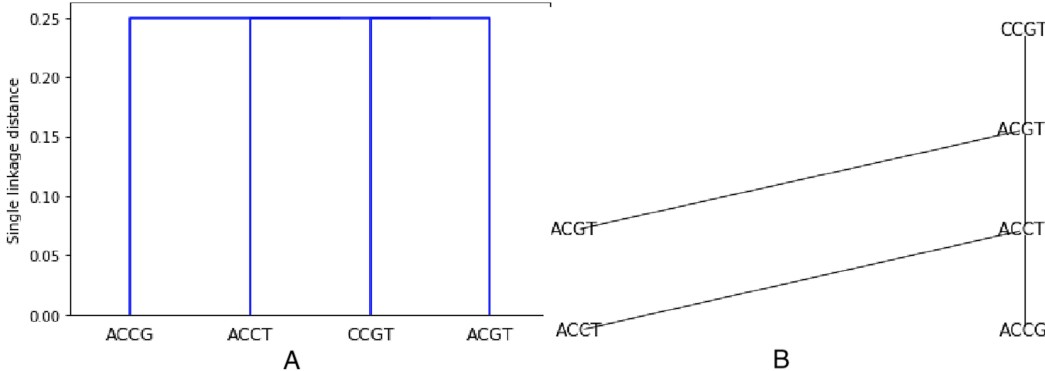

**Figure 3 Hierarchical clustering the set S = {ACGT, ACCT, ACCG, CCGT} with respect to the Hamming distance and using single linkage, resulting in a single linkage distance of 0.25 in every case.**

clustering starts from one cluster and iterates top down). This process is formalized in Algorithm 2.

While there are more than one clusters, a distance matrix $D$ containing between-cluster distances is computed (there are different possibilities for what the between-cluster distance $d_{BC}$ should be, it is also referred to as 'linkage' distance). The two closest clusters as per $d_{DB}$ are agglomerated, and the individual clusters are discarded. For computational efficiency, one would update the matrix $D$ by computing the between-cluster distance between existing clusters and the newly created one, and delete the obsolete rows rather than recompute the whole matrix.

An example of between-cluster distance is single linkage, given by:

$$d_{BC}(C_i, C_j) = \min_{x \in C_i, y \in C_j} d(x, y),$$

Meaning that the smallest pairwise distance between elements of $C_i$ and $C_j$ becomes the distance between the two clusters.

As an example, the set $S$ from previous examples (see Figs. 1 and 2) is clustered using the Hamming distance $d = d_H$ and single linkage. The resulting dendogram is shown on Fig. 3A. It is easy to understand since we already know that the minimum spanning tree is made of only edges with weight 1. Thus two closest sequences are first combined (there are three choices: (ACCT, ACCG), (ACCT, ACGT), (ACGT, CCGT)), they are at distance one from each other. But then, every sequence not already chosen will be at distance 1 from one sequence in the chosen set. Thus all sequences are at the same level, namely at distance $1/4$ in the dendogram (the dendogram distance of 1 is normalized by the length of 4).

We observe that:

- A dendogram resulting from clustering gives a tree structure, not a labeling of this tree.
- Once the dendogram and thus the tree structure is obtained, several labelings are possible, this is in effect the small parsimony problem defined in "Formal definitions".

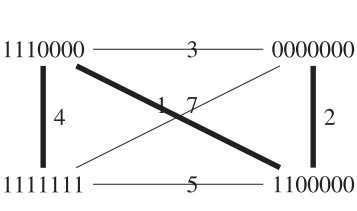

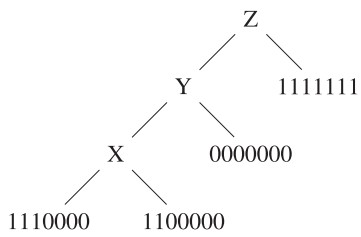

**(a)** The minimum spanning tree of $G(S)$.

**(b)** The corresponding dendogram tree structure with labels.

**Figure 4** **The set $S = \{1111111, 1110000, 1100000, 0000000\}$ with its the corresponding graph $G(S)$ and dendogram.**

We propose Algorithm 3 in which we modify the single linkage agglomerative clustering to include a labeling which comprises only points to be clustered.

The leaves are labeled by the points. Then the labels are attributed after the next level of agglomeration is decided, and the points that decide the agglomeration are *a posteriori* used to label the intermediate nodes. By construction, the output tree $T$ has labels all in $S = \{P_1, \ldots, P_n\}$. This is illustrated on Fig. 3B.

Connections between minimum spanning trees and single linkage clustering are well known (*Gower & Ross, 1969*). The knowledge of a minimum spanning tree is enough to construct clusters from a single linkage hierarchical clustering (*Gower & Ross, 1969*): group all points into disjoint sets by joining all dendogram segments of length less or equal to a given threshold. This will give clusters, which can be obtained from a minimum spanning tree by removing all edges of weight more than the chosen threshold. Thus the above algorithm constructs a minimum spanning tree, and also a dendogram, whose clusters are in correspondence with the obtained minimum spanning tree. This provides a solution to the large parsimony problem, in that it fixes a tree structure, however there is no guarantee on the weight of the resulting dendogram, since there is no guarantee on the optimality of the labeling, even though they are restricted in $S$. Furthermore, for better parsimony, the labeling could be optimized by searching for labels outside $S$.

We show next that the dendogram structure obtained from the single linkage hierarchical clustering may in fact have no labeling with weight smaller or equal than that of a minimum spanning tree.

**Proposition 2.** *There exist sequence sets S whose single linkage dendograms have no labeling with weight smaller or equal than that of a minimum spanning tree of G(S).*

*Proof.* Consider the set $S = \{1111111, 1110000, 1100000, 0000000\}$. Its complete graph, minimum spanning tree and dendogram are shown on Figs. 4A and 4B. By construction, the dendogram starts with grouping 1110000 and 1100000 which are at distance 1 from each other. Any intermediate label $X$ brings a weight of

$$d_H(1110000, X) + d_H(X, 1100000).$$

At the second iteration of the dendogram construction, 0000000 is added, because it is at distance 2 from 1100000. Then again, any intermediate label $Y$ brings a weight of

$$d_H(X, Y) + d_H(Y, 0000000).$$

Similarly, for $Z$, any intermediate label adds a further weight of

$$d_H(Y, Z) + d_H(Z, 1111111).$$

Any labeling $X$, $Y$, $Z$ thus yields for the obtained dendogram a weight of

$$d_H(1110000, X)$$
$$+ d_H(X, 1100000) + d_H(X, Y) + d_H(Y, 0000000) + d_H(Y, Z) + d_H(Z, 1111111)$$
$$\geq d_H(1110000, X) + d_H(X, 1100000) + d_H(X, Y) + d_H(Y, 0000000) + d_H(Y, 1111111)$$
$$= d_H(1110000, X) + d_H(X, 1100000) + d_H(X, Y) + 7$$

using first the triangle inequality, and then that

$$d_H(Y, 1111111) = 7 - d_H(Y, 0000000)$$

since the left hand-side term counts the number of coordinates of $Y$ that are 0, the right hand-side term removes the number of coordinates that are one from seven. Since the weight of the minimum spanning tree is 7, any labeling will give a weight strictly larger than 7 if and only if

$$d_H(1110000, X) + d_H(X, 1100000) + d_H(X, Y) > 0.$$

This is always the case, because in order to have equality, we would need all the three distances to be 0, which is impossible.

The example given in the above proof is not isolated. Suppose a sequence $s_1$ is far away from all other sequences in $S$ (it is the case of $s_1 = 1111111$ in the proof). When the sequence $s_1$ and the lastly added sequence $s_2$ form an edge $(s_1, s_2) \in G(S)$ which not only is not in a minimum spanning tree of $G(S)$, but in fact has weight large enough that it cannot be compensated ($s_2 = 0000000$ in the proof), even if there were weights less than those of the minimum spanning tree added before, then no labeling exist. The proof provides an extreme example where $d_H(s_1, s_2) = n$ which is the length of the sequences, thus the largest possible distance, and also in this case the weight of the minimum spanning tree.

An insight that could be learned from the above proof is that the weight of the tree can be explicitly written in terms of distances between leaves and intermediate nodes, or between two intermediate nodes. Minimizing the weight globally forces the sum of these distances to be reduced. By the triangle inequality, this means that upper bounds on the distances between leaves is also globally reduced. However, going through intermediate nodes adds constraints that are not there when only considering distances among leaves, which is the case in a hierarchical clustering.

Let us compare Algorithms 1 and 3. The first algorithm shows that it is always possible to build a tree with weight that of a minimum spanning tree (and the bound of Proposition 1 tells us that a better weight could be possible). The second algorithm provides a tree whose structure is that of a single linkage hierarchical clustering dendogram, however the

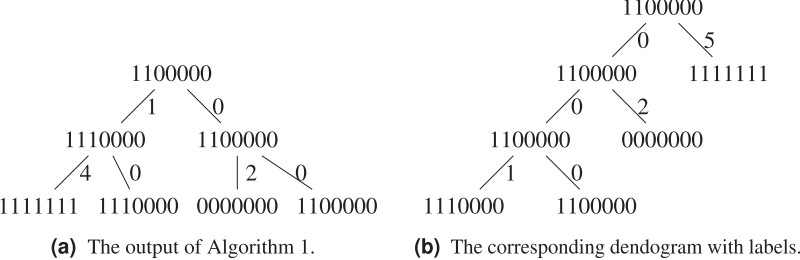

**(a)** The output of Algorithm 1.                    **(b)** The corresponding dendogram with labels.

**Figure 5 Comparison between the output of Algorithm 1 and Algorithm 3.**

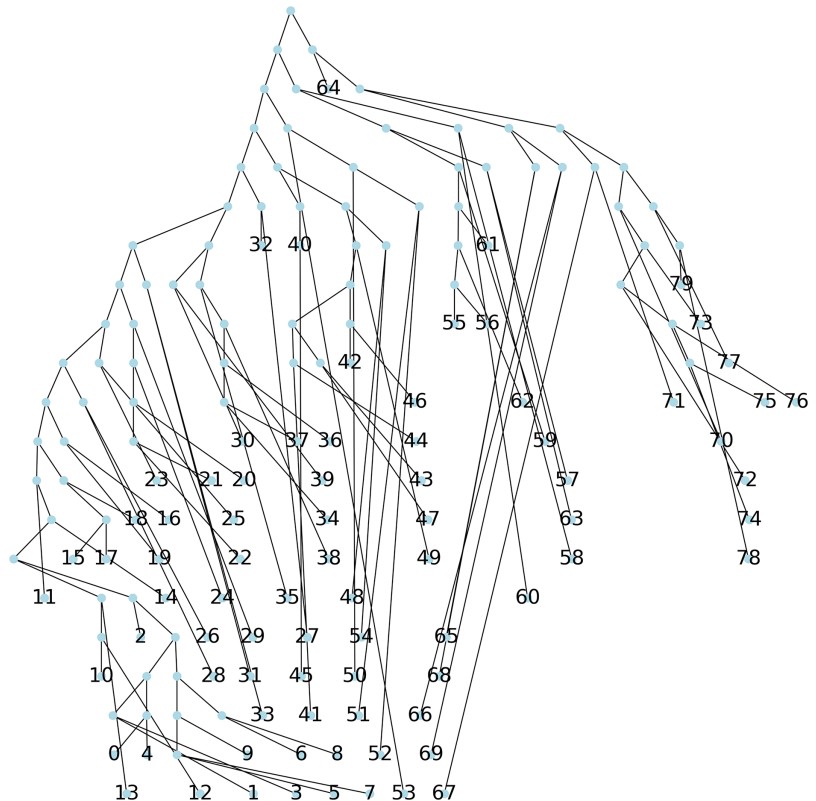

**Figure 6 The parsimonious tree constructed *via* the NNITreeSearcher by the ParsimonyTreeConstructor of Biopython from an initial tree found by the neighbor joining algorithm.**

example in the above proof illustrates that there are sequences for which this tree structure will give a weight strictly worse than that of a minimum spanning tree.

Let us exemplify the difference using the above example, see Figs. 5A and 5B. Algorithm 1 groups 1111111 and 1110000 in the same cluster, while both of them will only find themselves in the same cluster at the last step of Algorithm 3, when all sequences form a single cluster. This is because the hierarchical clustering algorithm starts with a local optimization, which groups first sequences that are the closest. The sequence 1111111 is further away from other sequences, it will be added last. Both algorithms agree on the fact that the ancestor of 1100000 and 0000000 is 1100000.

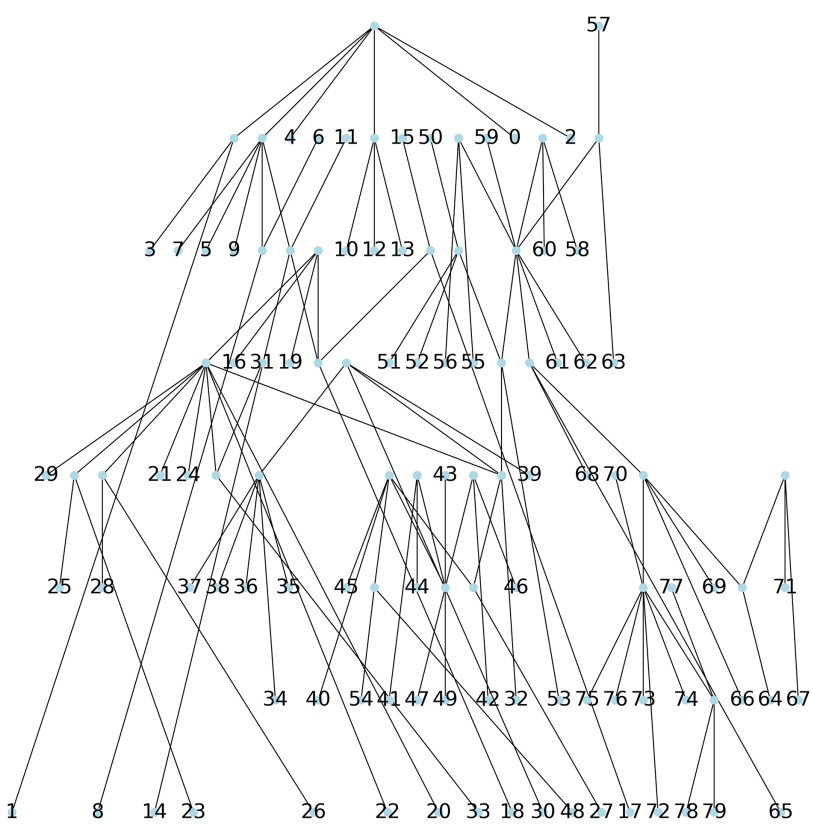

**Figure 7 The parsimonious tree obtained from Algorithm 1.**

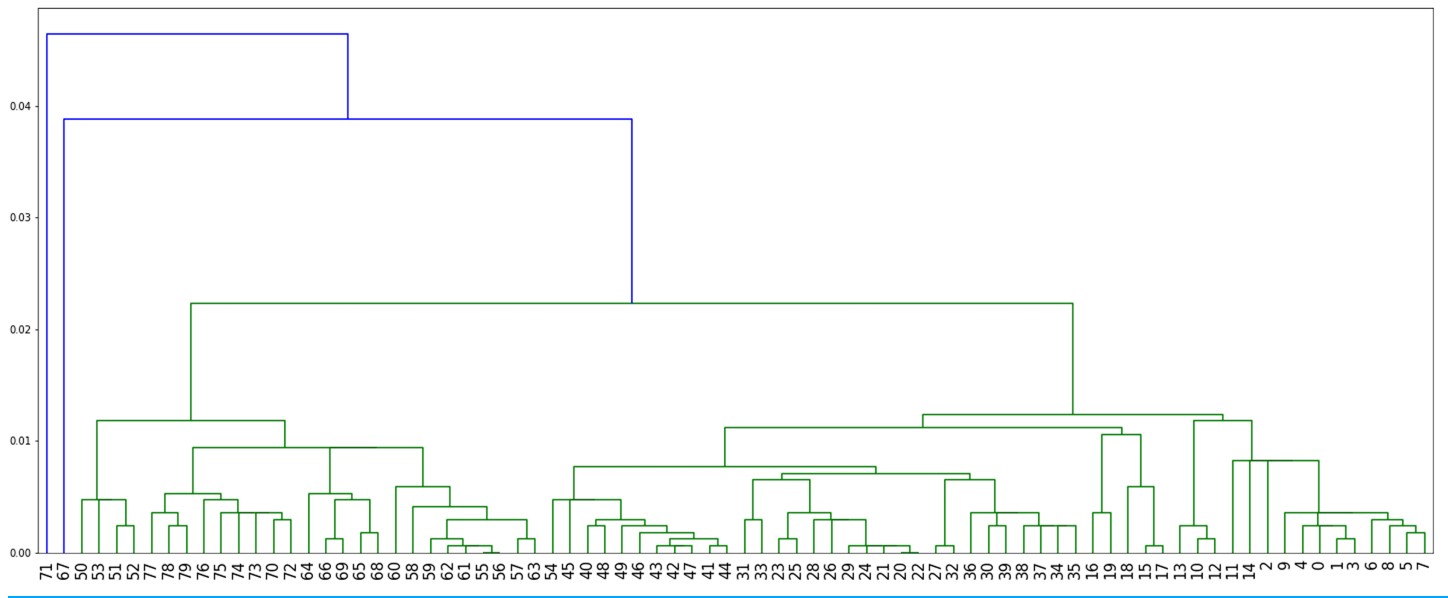

**Figure 8 The dendogram resulting from the clustering of the US flu dataset with respect to the Hamming distance and using single linkage.**

**Table 1 F-scores for clusters obtained through four different methods.**

**(A) Comparisons between 4, 4, 6 and 4 clusters**

|           | $C_1(4)$ | $C_2(4)$ | $C_3(6)$ | $C_4(4)$ |
|-----------|----------|----------|----------|----------|
| $C_1(4)$  | 1        | 0.8566   | 0.7006   | 0.7736   |
| $C_2(4)$  |          | 1        | 0.5857   | 0.8579   |
| $C_3(6)$  |          |          | 1        | 0.6010   |
| $C_4(6)$  |          |          |          | 1        |

**(B) Comparisons between 6, 7, 6 and 8 clusters**

|           | $C_1(6)$ | $C_2(7)$ | $C_3(6)$ | $C_4(8)$ |
|-----------|----------|----------|----------|----------|
| $C_1(6)$  | 1        |          |          |          |
| $C_2(7)$  | 0.9177   | 1        |          |          |
| $C_3(6)$  | 0.5655   | 0.5076   | 1        |          |
| $C_4(8)$  | 0.5333   | 0.5666   | 0.4513   | 1        |

## NUMERICAL EXPLORATION

For validating the algorithms, and getting some sense of the quality of the obtained clusterings, we explore a dataset $S$ of 80 DNA sequences for different influenza strains collected from 1993 to 2008 in the US (available at *adegenet (2023)*). Each sequence in $S$ has the length of 1,701.

We consider four methods, Methods 1 and 2 rely on Biopython (*Cock et al., 2009*), while Methods 3 and 4 are derived from the proposed algorithms. Biopython provides a tree constructor, ParsimonyTreeConstructor, which takes as input a searcher (NNITTreeSearcher, see *Robinson (1971)* for the Nearest Neighbor Interchanges (NNI) algorithm), and optionally an initial tree.

- **Method 1**: a parsimonious tree is constructed using Biopython, and no initial tree. This initial tree is thus computed by default. This results in a tree with a parsimony length of 543.
- **Method 2**: a parsimonious tree is obtained using Biopython from an initial tree found through the Neighbor Joining algorithm (NJ) (*Saitou & Nei, 1987*), resulting in a parsimony length of 540, see Fig. 6.
- **Method 3**: we implemented Algorithm 1 which creates a parsimony tree whose weight is that of a minimum spanning tree for $G(S)$, namely 671, see Fig. 7.
- **Method 4**: we implemented Algorithm 3 which assigns labels to minimize parsimony on the dendogram tree obtained from single linkage hierarchical clustering in python, relying on the libraries networkx (*Hagberg & Conway, 2020*) and scipy (*Virtanen et al. 2020*). The corresponding dendogram is shown on Fig. 8. It has weight 988.

We note here that when comparing the obtained parsimony lengths or weights, namely 543, 540, 671 and 988, it should be remembered that the sequences have length 1,701.

While it may be possible to look at the three plots in Figs. 6–8 to identify patterns, such visual comparison is complicated and difficult to quantify similarities and differences. As such, we use the F-score of two clusters (*Pfitzner, Leibbrandt & Powers, 2009*) as a quantitative way to compare the resulting clusterings. The F-score applies even if different numbers of clusters are available.

The pairwise F-score of the resulting clusters are summarized in Table 1. We use the following notations:

- $C_1$, $C_2$, $C_3$ and $C_4$ respectively corresponds to Method 1 (a parsimonious tree obtained from scratch and optimized with NNI), Method 2 (the parsimonious tree obtained from the NJ algorithm), Method 3 (the tree obtained from Algorithm 1) and Method 4 (the tree obtained from Algorithm 3).
- The index $k$ in $C_i(k)$ for $i = 1, 2, 3, 4$ refers to the number of clusters considered. Clusters are obtained by truncating each tree at different depths from the root. Since trees have different structures, it may not be possible to obtain the same number of clusters.

  We make the following observations: (a) clusters obtained using NNI and NJ algorithms for parsimony tree construction are relatively closer to each other across the different granularities of clustering studied; (b) the MST based (Algorithm 1) approach results in the most distinct clusters; (c) the hierarchical Algorithm 3 yields something intermediate and closer to the clusters obtained with NJ algorithm; (d) overall, scores are lower in Table 1B compared to Table 1A, at the exception of $C_1(7), C_2(7)$.

This illustrates divergent patterns between parsimony behaviour and tree structures. The parsimony is highest with Method 4 (980), and then decreases through Method 3 (671), Method 1 (543) and is the lowest with Method 2 (541). However, the clusters obtained from Method 3 give the lowest F-scores and are thus most distinct from the others.

## CONCLUDING REMARKS

This work looked at the cluster structure obtained from parsimonious trees, using Fitch's parsimony, in particular in comparison with the clusters obtained from hierarchical clustering. While it gave a first glance into the problem, particularly, Proposition 2 establishes a clear demarcation: optimal parsimonious trees have parsimony less than or equal to the weight of the minimum spanning tree, while there exists datasets where single linkage hierarchical clustering would necessarily lead to tree structures whose parsimony would be larger, many questions remain, in terms of understanding the tree structure that parsimony as a criterion is trying to optimize. It could be interesting to categorize datasets and possibly find some for which this demarcation is different. Also, while single linkage is one form of linkage that looked the most natural to consider, another forms of linkage could be studied. Alternatively, single linkage hierarchical clustering could be used as a seed for further heuristics. Further extensions include Wagner parsimony and distance-based methods. For instance, for the case of Wagner's method, a software such as TNT

(*Goloboff, Farris & Nixon, 2008*) could be used to obtain comparison results. A different direction would be to compare forests of cladograms *vs*. forests of dendogram instead.

## ACKNOWLEDGEMENTS

We would like to thank Daubian Santos, for his suggestions for further discussions that have been incorporated in the concluding remarks.

### Funding

The authors received no funding for this work.

### Competing Interests

Anwitaman Datta is an Academic Editor for PeerJ Computer Science. Frédérique Oggier has no competing interests.

### Author Contributions

- Frédérique Oggier conceived and designed the experiments, performed the experiments, analyzed the data, performed the computation work, prepared figures and/or tables, authored or reviewed drafts of the article, and approved the final draft.
- Anwitaman Datta conceived and designed the experiments, analyzed the data, prepared figures and/or tables, authored or reviewed drafts of the article, and approved the final draft.

### Data Availability

The code is available in the Supplemental File.

The data is available at adegenet: usflu fasta, http://adegenet.r-forge.r-project.org/files/usflu.fasta.

### Supplemental Information

Supplemental information for this article can be found online at http://dx.doi.org/10.7717/peerj-cs.1339#supplemental-information.

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
