# Peer review of "On parsimony and clustering"

_PeerJ Computer Science, doi:10.7717/peerj-cs.1339_

## Round 0.1 · original submission · Major Revisions

The reviewers found this article interesting but raised a number of points that need to be fixed in this study, which cannot be accepted for publication in its current status. The authors should carefully study the reviwers' reports and prepare a new version of the manuscript that addresses them.

·

Basic reporting

The text is well-written, objective and informative. The introduction is a good example of an excellent and concise contextualization. The problem is clear in the text, and it is relevant. In addition, the authors may cite some potential datasets to clarify the utility of the issue. There is a good use of figures. The text is self-contained. In a brief commentary about the contextualization, the authors highlighted the parsimony approach of mutant viruses and lineages. Although it is true that the most famous one is about HIV, I guess that you find more recent and useful citations in COVID-19 studies. Another formality, the authors used in the text was a weird citation mode. For example: "were addressed in (Brower, 2000)". The most usual way would be: "were addressed in Brower (2000)". I do not know if it is a specific rule of the journal or a stylistic choice of the authors. Also, in the references, we miss the direct citation of Hennig (1950) and Edwards and Cavalli-Sforza (1963).

Experimental design

The algorithms and methods are interesting and meaningful. However, I think that the result, although relevant, it is a partial answer to the question. Wagner's method, for example, is easy to compare. There is some software such as TNT that are frequently used parsimony and may be good comparison results. Maybe, the more practical solution is adequate for the question to explore a more reduced scope (and the word "connection" may be useful 'with parsimony' in a dendrogram context). Instead of "what are the connections, if any, between the cladograms obtained using maximum parsimony as an optimization criterion, and dendograms obtained from agglomerative hierarchical clustering?" it may be "what are the congruences, if any, between the cladograms obtained using Fitch parsimony as an optimization criterion, and dendograms obtained from agglomerative hierarchical clustering?".

Validity of the findings

I still feel a necessity to explore more a discussion of the results. The resemblances are better in large or small parsimony? The level of approach is sufficient? Partially sufficient? Is enough level of similarity? It can be measured? Besides that, it is these methods strictly divergent or it is possible for the clustering approach to work as the first seed of heuristic methods? Although there is a page limit, a more detailed discussion of the results will enrich substantially the study. even that, the text is a well-analyzed test about a specific problem with many interesting echoing future studies.

Additional comments

Double-checked the submitted figure 3 (tody.png) because it may be an incomplete file.

Reviewer 2 ·

Basic reporting

See comments below

Experimental design

See comments below

Validity of the findings

See comments below

Additional comments

Put very simply, Oggier and Datta compare minimum spanning trees, hierarchical clustering trees, and most parsimonious trees. The hope seems to be that agglomerative clustering might provide a useful approximation to a parsimony tree, though the authors do not give any indication of how a method might be “validated” as useful (a certain tree distance? Within a certain percentage of the optimal tree score?).

The single dataset that the authors analyse is sufficient to show that the method is not even close to being useful, and thus it is very unlikely that a more robust exploration of the method (e.g. evaluating enough datasets to indicate that a result might be generalizable) is worthwhile. (Trees with scores even a couple of steps away from a global optimum are often very different from the most parsimonious tree; the authors’ algorithms return scores >20% less parsimonious than the best scores found).

If the methods have any value, perhaps they could compete with other methods for creating “starting trees” for use as seeds for parsimony search, such as neighbour joining, adding taxa in turn to the most parsimonious location on a growing tree, or even random addition. But the authors do not even demonstrate that their method is competitive with the computationally cheap neighbour-joining approach.

As it stands, I don’t see anything in the paper that is surprising or likely to be useful in practice, so I am not recommending the paper for acceptance.





L94. Perhaps clarify that “all sequences” means “all possible sequences” (not “all observed sequences”)? Please clarify what is meant by “typically”; the only other case I can think of is a single-member alphabet, reflecting invariant characters. I am also unclear whether a single alphabet is necessary for all sequences; e.g. a discrete dataset may contain some characters (/sites / position) that employ the tokens 0/1 for “present / absent”, and other characters that employ the tokens 1/2/3 for “red/green/blue” (say). Are additive characters (i.e. where 0→2 has a weight of 2, but 0→1 and 1→2 each have a cost of 1) considered?

L103: I don’t at this juncture see the point of the term “parsimonious tree”; I am missing how the definition in L103 distinguishes a “parsimonious tree” from a “tree”? Parsimony is a property of a tree, so whilst some trees can be more or less parsimonious, all trees have a length (i.e. parsimony “score”). Because the term “unparsimonious” is sometimes used as a synonym for “not most parsimonious”, readers may assume that “parsimonious tree” is intended to mean “most parsimonious tree”. Would the term “parsimoniously labelled tree” better express the intent? (Notwithstanding my comment on L111…)

L111. Not clear from the text that the “optimal labels” refers to a label set for the internal nodes, i.e. that the leaf labels are given with the tree. Suggest rewording along the lines of “finding the optimal labels for internal nodes given a tree with labelled leaves”.
L112. I think “addressed” is intended to mean “solved”; please make clear.

L117. Should “The set of sequences” read “a set of sequences”, or “the set of sequences for a given selection of leaves”?

L130. Please check that this is an accurate summary of Jones et al.; or clarify the nature of the fixed threshold (my first reading of the written text was that any fixed threshold could be selected, whereas my impression from Jones et al. is that the threshold is a function of tree size)

L152. If this sentence is not relevant, why include it?

Line 223; caption to fig. 6; elsewhere?: Trees with a smaller parsimony score are more parsimonious, so this tree is (presumably) obtained by maximizing, not minimizing, parsimony.

Line 223. Tree search does not optimize a tree. I think the process described here is “A tree was selected at random and used as a starting tree for maximum parsimony search using only NNI rearrangements”. If so, then the score of 543 reflects the score of a single iteration of this process; had a different starting tree been selected, or different NNI rearrangements employed, a different length may have been obtained. The authors should specify the conditions under which tree search was terminated: would running the search for longer have found shorter trees? If the purpose of this method is to approximate the optimal parsimony score, then why are other approaches, such as TBR rearrangements or the parsimony ratchet, not employed? Heuristic search that rapidly finds a globally optimal score can be conducted in moments on a small dataset such as this one in software such as TNT.

Line 226. What method was used to “optimize the tree”? If the same process described in method 1 was employed, this should be stated; or do you simply mean that internal nodes were labelled parsimoniously? Surely the (unoptimized) NJ tree would be a more relevant point of comparison to methods 3 & 4?

Line 237. The sequence length goes some way to contextualizing these values; more useful would be the expected parsimony score of a randomly selected parsimonious tree.

L240. The question stated in L57 requires a comparison of cladograms (not their clusterings). Would the clustering information distance (Smith 2020, Bioinformatics) offer a complementary and perhaps more complete view of the similarity of trees (and their clusterings)?

L244. Would giving the methods more memorable names avoid having to redefine them here?

Caption to Fig. 6. The caption does not stand alone, and I cannot infer what is meant without reference to the text.

Figs 6 & 7 would be easier to read if the edges did not cross. What is the rationale for the x and y coordinates of the points?

Cite this review as

·

Basic reporting

no comment

Experimental design

The figures 6 and 7 should be enhanced. Please use another package to plot them, maybe the phylo package.

Validity of the findings

no comment

Additional comments

The article at hand revisits parsimonious cladograms through the lens of clustering and
compares cladograms optimized for parsimony with dendograms obtained from single linkage hierarchical clustering. The text is well structured and clear.
Here is one comment that needs to be addressed: The authors should make the answer to their research question more obvious in the CONCLUDING REMARKS section. That would enhance the readability of their article.

---

## Round 0.2 · Minor Revisions

The authors addressed most of the points raised by the reviewers in a clear way. Just some very minor changes are necessary before a potential acceptance:
1) Please replace all the web URLs in the text with references to those web URLs.
2) Please replace all the occurrences of "e.g." with "for example" or "for instance"

Thanks

·

Basic reporting

The authors answer my questions and rewrite sufficiently

Experimental design

The authors answer my questions and rewrite sufficiently

Validity of the findings

The authors answer my questions and rewrite sufficiently

·

Basic reporting

See comment below

Experimental design

See comment below

Validity of the findings

See comment below

Additional comments

The authors did a good job addressing all the previous comments and clarifying the changes they made to enhance the manuscript text. The revised manuscript seems in a good shape for publications.

---

## Round 0.3 · accepted · Accept

The authors addressed all the points raised by the reviewers and therefore I recommend this article for acceptance.